# The Product of the Fission Yeast *fhl1* Gene Binds to the HomolE Box and Activates In Vitro Transcription of Ribosomal Protein Genes

**DOI:** 10.3390/ijms24119472

**Published:** 2023-05-30

**Authors:** Edio Maldonado, Sebastian Morales-Pison, Fabiola Urbina, Claudia Arias, Christian Castillo, Lilian Jara, Aldo Solari

**Affiliations:** 1Programa de Biología Celular y Molecular, Instituto de Ciencias Biomédicas, Facultad de Medicina, Universidad de Chile, Santiago 8380453, Chile; fabi.urbina1516@gmail.com (F.U.); agathasandiego@gmail.com (C.A.); ccastillor@uchile.cl (C.C.); 2Centro de Oncología de Precisión (COP), Facultad de Medicina y Ciencias de la Salud, Universidad Mayor, Santiago 7560908, Chile; sebastian.morales@umayor.cl; 3Facultad de Medicina Veterinaria y Agronomía, Universidad de las Américas, Santiago 7500975, Chile; 4Programa de Genética Humana, Instituto de Ciencias Biomédicas, Facultad de Medicina, Universidad de Chile, Santiago 8380453, Chile; ljara@uchile.cl

**Keywords:** HomolE box, Fhl1 protein, HomolD box, RPGs transcription

## Abstract

Fission yeast ribosomal protein genes (RPGs) contain a HomolD box as a core promoter element required for transcription. Some of the RPGs also contain a consensus sequence named HomolE, located upstream of the HomolD box. The HomolE box acts as an upstream activating sequence (UAS), and it is able to activate transcription in RPG promoters containing a HomolD box. In this work, we identified a HomolE-binding protein (HEBP) as a polypeptide of 100 kDa, which was able to bind to the HomolE box in a Southwestern blot assay. The features of this polypeptide were similar to the product of the *fhl1* gene of fission yeast. The Fhl1 protein is the homolog of the FHL1 protein of budding yeast and possesses fork-head-associated (FHA) and fork-head (FH) domains. The product of the *fhl1* gene was expressed and purified from bacteria, and it was demonstrated that is able to bind the HomolE box in an electrophoretic mobility assay (EMSA), as well as being able to activate in vitro transcription from an RPG gene promoter containing HomolE boxes upstream of the HomolD box. These results indicate that the product of the *fhl1* gene of fission yeast can bind to the HomolE box, and it activates the transcription of RPGs.

## 1. Introduction

The protein-synthesizing machinery in all living cells is the ribosome. The ribosome is composed of 55 ribosomal proteins (RPs) and 3 ribosomal RNAs (rRNAs) in *Escherichia coli*; in mammals, it is composed of 80 RPs and 4 rRNAs, while in budding yeast it is composed of 79 RPs and 4 rRNAs [1,2]. The transcription of ribosomal protein genes (RPGs) in all species is thought to be tightly controlled and coordinately expressed. In eukaryotes, these genes form a transcriptional module containing one or more common DNA motifs, which bind transcriptional activators [3]. In *E. coli*, the RPGs form an operon in which a single promoter controls the expression; however, in eukaryotes—such as yeasts, plants, and animals—regulation is much more complicated and less understood than in prokaryotes [3]. In eukaryotes, the RPGs are scattered throughout the entire genome with no operon structure but, surprisingly, co-regulation and the coordinated expression are still observed [3,4]. Since this transcriptional module is small and the amino-acidic sequence of RPs is highly conserved, understanding the basic principles coordinating gene regulation of this transcriptional module will significantly contribute to the knowledge and function of other transcriptional modules.

The fission yeast genome contains 141 RPGs, encoding the complete set of 79 RPs, forming the ribosome with four rRNAs [5,6]. Since only 79 RPs form the ribosome and there are 141 RPGs, a single RP can be encoded for more than one gene, indicating that some RPGs have been duplicated and can constitute a family of RPGs [5,6]. Most RPs in plants and fungi are encoded for more than one RPG, while mammals and protists contain only a few duplicated RPGs [7]. The duplicated genes are thought to have arisen from genome duplication, genome hybridization, retroduplication and, in some cases, polyploidy. The differences in the numbers of RPGs in these genomes are not related to genome size but, rather, are associated with the organism’s overall propensity to gene duplication [8,9].

The initial characterization of 14 promoters of RPGs in fission yeast showed discrete conserved modules, which were named the homology (Homol) A, B, C, D, and E regions [5,10,11]. Surprisingly, these Homol regions were completely different from previously described promoter elements of budding yeast and mammals such as Rap1 (budding yeast), TATA box, initiator (Inr), downstream promoter element (DPE), TCT initiator, and motif 10 element (MTE). The function of each Homol region was studied using a promoter deletion approach in fission yeast. It was found that the HomolA, -B, and -C regions play a role in regulating transcription and might have an upstream activating sequence (UAS)-like function; moreover, they were not conserved in all of the RPG promoters [5,10,11]. In contrast, the HomolD sequence was present in all of the studied RPG promoters, and it was able to direct the initiation of transcription in an analog fashion, similar to the TATA box [5,10]. This conserved sequence is the octamer CAGTCACA/G and is located 39–52 base pairs (bp) upstream of the transcription start site (TSS), in a position where the TATA box is usually located in fission yeast [5,10]. In some promoters, the inverted form TGTGACTG is found and is fully functional to direct the initiation of transcription. In an in vivo assay, using a reporter gene assay, it was demonstrated that the HomolD box is necessary and sufficient to direct and initiate transcription from the RPGs and, thus, can act as a TATA box analog. An electrophoretic mobility shift assay (EMSA) also found that the HomolD box binds a factor different from the TATA-box-binding protein (TBP). Point mutations in the HomolD box can completely abolish the ability of this DNA motif to direct the initiation of transcription from the RPGs [5,10].

RPs from fission yeast can be encoded by two or three duplicated RPGs, which contain a HomolD box in their promoters to initiate transcription. Interestingly, in each RPG family, at least one of those members possesses a sequence upstream of the HomolD box named HomolE box, which is a tandem repeated DNA motif with a consensus sequence of ACCCTACCCT or the inverted form AGGGTAGGGT [5,11]. This DNA motif corresponds to a proximal UAS-like element for HomolD-containing promoters, since this sequence upstream of the HomolD box strongly increases transcription in vivo [5,11]. The HomolE and HomolD can occur in those promoters in both orientations, and the distance between them varies from 0 to 32 bp [5,11]. Five size classes of spacers can be found setting the HomolE–HomolD promoter arrangement, which are 0–2, 6–8, 10–12, 14–17, and 21 bp apart [5,11]. The distance between both boxes is critical in the transcriptional activation of RPGs. It has been demonstrated that the smaller the distance between the HomolE and HomolD boxes, the higher the activity of the promoter arrangement [5,11]. The sequence context between HomolE and HomolD is also important for transcriptional activation. To be fully functional, both promoter elements must be in the same orientation as they are in natural promoters. For example, the arrangement AGGGTAGGGT-TGTGACTG found in a natural promoter is fully functional and strongly activates transcription; in contrast, the inverted arrangement ACCCTACCCT-CAGTCACA only has basal activity [5,11].

The fission yeast *Schizosaccharomyces pombe* is a valuable biological model for studying cellular processes, and genetic, biochemical, and bioinformatic tools can be applied. In this work, we identified a HomolE-binding protein from fission yeast and found that the recombinant protein can activate in vitro transcription in a cell-free extract using a synthetic promoter containing a HomolD box and two upstream HomolE boxes. The identified HomolE-binding protein (HEBP) is the product of the *fhl1* gene.

## 2. Results

### 2.1. Identification of a HomolE-Box-Binding Protein in Fission Yeast Whole-Cell Extracts

A HomolE-box-binding factor has been previously identified in whole-cell extracts (WCEs) from fission yeast [11]. Using a similar EMSA assay, we tried to identify a similar binding factor in our laboratory. However, by using a double-stranded end-labeled oligonucleotide containing the HomolE box and calf thymus DNA as a non-specific competitor, we were not able to find stable complexes from fission yeast WCEs. Therefore, we used an end-labeled probe containing two HomolE boxes (see Section 4) and sonicated salmon sperm DNA as a non-specific competitor. Using that assay, a stable DNA–protein complex from WCEs was observed (Figure 1, lane 2). Figure 1 shows that the bound factor is specific to the HomolE box, since the binding could not be competed with a mutant HomolE box oligonucleotide (lane 3) or a HomolD box oligonucleotide (lane 4); however, it was competed with the wild-type HomolE box oligonucleotide (lanes 5–7). It was concluded that the HomolE box is the target of a protein factor present in fission yeast WCEs. Therefore, we decided to proceed further and purify the HomolE-box-binding protein (HEBP) using chromatographic techniques, in an effort to identify this protein.

### 2.2. Purification of the HEBP from Fission Yeast Whole-Cell Extracts

A fission yeast WCE was subjected to fractionation using different chromatographic resins (Figure 2A), and the HEBP activity was monitored by EMSA (see Section 4). The last step of purification was performed on a Poros-S resin. The binding activity peaked sharply from this last column and eluted around 0.25 M KCl concentration (Figure 2B, fractions 12–14). These fractions were analyzed by SDS-PAGE, followed by Coomassie Blue R-250 staining to analyze the polypeptide composition (Figure 2C). We observed a major polypeptide of around 100 kDa and several others of higher and lower molecular weights. We thought that the 100 kDa polypeptide might be a good candidate for the HEBP. Therefore, we decided to analyze that polypeptide further.

### 2.3. The 100 kDa Polypeptide Contains the HomolE-Box-Binding Activity

Southwestern blotting is a valuable tool to investigate DNA–protein interactions, since it has an advantage over the EMSA in providing information regarding the molecular weight of an unknown transcription factor. Thus, we analyzed the protein fraction containing binding activity towards the HomolE box from the Poros-S column by Southwestern blotting. The proteins were separated by SDS-PAGE and transferred to a membrane, and the membrane-bound proteins were renatured and incubated with the end-labeled double-stranded HomolE box probe used for the EMSA. The mutated end-labeled HomolE box probe was used as a negative control. Figure 3 (panel B) shows that the HomolE box probe interacts with the major 100 kDa polypeptide from the fraction. The minor polypeptides did not bind to the HomolE box probe. The binding of this polypeptide was specific to the wild-type probe, since it did not interact with the mutated end-labeled HomolE box (Figure 3, panel A). Therefore, we concluded that the 100 kDa polypeptide present in the Poros-S fraction was an HEBP.

### 2.4. The 100 kDa Polypeptide Might Be the Product of the Fork-Head-Like 1 (fhl1) Gene

We thought that the 100 kDa polypeptide might be the product of the *fhl1* gene, since this gene encodes a 723-amino-acid polypeptide with a predicted molecular weight of 81.13 kDa. This is consistent with the molecular weight calculated by Southwestern blot. Furthermore, two additional observations support this idea: First, the homolog gene from budding yeast (*FHL1*) is involved in RPG transcription [12]. The IFHL-binding motif is TC(C/T)GCCTA. It has been demonstrated that the FHL1 factor from budding yeast binds to the IFHL motif in the RPG promoters [12]. Second, the HomolE box is a duplicated variant of the CCCTA motif, which resembles the second part of the IHFL motif GCCTA [13]. Moreover, the promoters of the RPG genes from the evolutionarily distant *Neurospora crassa* and *Aspergillus nidulans* contain a DNA motif (GCCCTA) that matches the half of the HomolE box [13]. Therefore, it is possible that the HomolE-binding protein belongs to the FHL family of transcription factors.

The fission yeast Fhl1 protein has close homologs in other fungal genera, such as *Schizosaccaromyces*, *Pneumocystis*, and *Smittium* (Appendix A). Like the budding yeast FHL1 factor, the fission yeast Fhl1 protein has FHA and FH domains, which are probably necessary for the transcriptional activity (Figure 4). The presence of the FH domain in the Fhl1 protein indicates that it most likely binds to the IFHL motif. On the hand, the FHA domain could interact with other transcription factors.

### 2.5. Recombinant fhl1 Protein Binds to the HomolE Box

To investigate whether the product of the *fhl1* gene binds to the HomolE box, we cloned the coding sequence of the *fhl1* gene in an expression vector to express and purify the Fhl1 protein. The protein was expressed, purified from bacteria, and tested in an EMSA assay. Figure 5A shows that the recombinant Fhl1 protein is more than 95% pure, as judged by SDS-PAGE, followed by silver staining. When the pure recombinant Fhl1 protein was assayed in EMSA, it formed a complex with the HomolE-box-labeled probe (Figure 5B, lanes 2 and 8). The binding was not completed by the cold mutant HomolE box oligonucleotide (lane 3) or cold TATA box oligonucleotide (Figure 5B, lane 4). However, the binding was completed by the cold HomolE box oligonucleotide (Figure 5B, lanes 5–7). Moreover, the binding was not completed using a cold HomolD oligonucleotide (Figure 5B, lane 9). These results indicate that the Fhl1 protein binds to the HomolE box in a specific fashion.

### 2.6. Recombinant fhl1 Activates Transcription from a Promoter Containing HomolE Boxes

Since the Fhl1 transcription factor binds to the HomolE box, we next investigated whether the recombinant Fhl1 protein was able to activate transcription from a synthetic promoter construct containing two HomolE boxes upstream of a HomolD box. Figure 6 shows that a promoter containing a mutated HomolE box was not activated by recombinant Fhl1 protein (compare lane 1 with lanes 2 and 3). In contrast, the HomolE-containing template was activated by recombinant Fhl1 protein (compare lanes 4–6 with lane 7). These results indicate that Fhl1 binds to the HomolE box, and it is able to activate transcription. Figure 6B shows the quantification of the results from Figure 6A, and it can be observed that there is a 4–6-fold activation with 40 ng of recombinant Fhl1 protein. These results indicate that the Fhl1 protein was able to activate in vitro transcription from a synthetic promoter containing HomolE boxes upstream of the HomolD box in a fission yeast WCE.

## 3. Discussion

In this work, we identified an HEBP from fission yeast as the product of the *fhl1* gene. The purified recombinant protein bound to the HomolE box and was able to activate transcription in a fission yeast WCE with a synthetic promoter containing both a HomolD box and upstream HomolE boxes, strongly suggesting that the product of the *fhl1* gene is an HEBP. The HomolE box is always found upstream of HomolD-box-containing promoters, which are present in RPGs and several non-RPGs.

Ribosome biogenesis is one of the most energy-intensive anabolic processes carried out by growing cells and is highly regulated in response to growth and stress signals [14,15,16]. It is estimated that in rapidly growing budding yeast, the ribosomes are produced at a rate of 2000 per minute and can be present at 200,000 copies per cell [16]. This requires the dedication of almost 50% of all RNA polymerase II (RNAPII) initiation events in RPG transcription and ribosome biogenesis genes (RiBi genes), along with a high production of rRNAs, which are produced by the combined action of RNA polymerase I (RNAPI) and RNA polymerase III (RNAPIII) [15,16]. The production of RPs and rRNAs must be highly coordinated in order to ensure a rapid and efficient ribosome assembly. This coordination must begin at the transcription level, stimulated by environmental signals that favor rapid growth and are downregulated by stress conditions. However, RPGs’ coordinated regulation and expression with other ribosomal components are still poorly understood in fission yeast. DNA microarray studies have shown that all 141 RPGs have tightly coordinated expressions of those genes. For example, during the switch from vegetative to meiotic growth, the expression of the RPGs is downregulated; however, within a short time, they are strongly reactivated at the beginning of meiosis [5,17]. The co-expression profile can be seen in 32 of the 59 non-RPGs containing a HomolD box, indicating that this DNA motif is responsible for the coordinated expression of RPGs and non-RPGs displaying a HomolD box in their promoters [5,17].

As stated earlier, the HomolE box has always been found upstream of the HomolD box in RPGs and in non-RPGs, which are transcribed by RNAPII [18]. The HomolD box binds a protein factor with biochemical features that are different from those of TBP. In an attempt to biochemically analyze the factor that binds to the HomolD box, we used DNA affinity chromatography linking a double-stranded multimerized HomolD sequence to the matrix to isolate the factor from a partially purified fraction of cell extract enriched in HomolD-binding activity [18]. Bound proteins were eluted from the affinity column and analyzed by mass spectrometry. The results indicated that the RNAPI transcription factor Rrn7 was identified in the bound fraction from the HomolD-box affinity column, which is required for RPG transcription [18]. The Rrn7 transcription factor is also a member of the RNAPI transcription machinery and is involved in the transcription of rRNA [18,19]. This factor binds to a conserved sequence in the rRNA gene, similar to the HomolD box [18,19]. However, the RPGs are transcribed by the RNAPII transcription apparatus, since the transcription in vitro is sensitive to α-amanitin [18]. This factor is a member of the Zn-ribbon protein family related to the RNAPII transcription factor TFIIB, which plays a pivotal role in the preinitiation complex (PIC) assembly on eukaryotic protein-coding genes [20,21]. Rrn7 possesses a Zn-ribbon at the N-terminus of the polypeptide and two cyclin fold domains at the C-terminus of the protein, and it displays domain conservation with the TFIIB family members [18,19,20,21]. The Rrn7 is required for transcription of the RPGs’ HomolD-box-containing promoters and, most likely, for the non-RPGs containing a HomolD box as well. This transcription factor is encoded by an essential gene for viability (Pombase), and unfortunately, no temperature-sensitive alleles have been isolated to study its in vivo function deeply. The HEBP (Fhl1 protein) probably interacts with Rrn7 to activate transcription or, alternatively, the Fhl1 protein interacts with the mediator complex or other components of the RNAPII transcription machinery.

Despite the conservation of the coding sequences of the RPGs from different eukaryotes, the promoter sequences controlling their expression are not conserved, and different cis-elements can be found in the different species. Therefore, it is necessary to compare the RPG promoters from different genomes and identify conserved cis-elements associated with them in each species. It is also necessary to identify the trans-acting protein factors that bind to each conserved DNA motif in the RPG promoters, and to study the molecular mechanisms associated with their expression. The transcriptional control of the RPGs is well known in budding yeast but differs from the transcriptional control of RPGs in fission yeast and mammalian cells [3,6,16]. In budding yeast cells, three distinct RPG promoter architectures lead to an extensive—albeit incomplete—transcriptional co-regulation [16]. Those promoter architectures have been named Category I, II, and III. Most of the promoters in budding yeast are bound by the Rap1 protein (127 out of 138), and this factor can be found in promoters of Category I and Category II promoters [16]. However, Rap1 also binds to a large number of non-RPG promoters [22]. Furthermore, the transcription factor FHL1 can interact and recruit IFH1 (interacts with fork-head 1); both are found at RPG promoters in budding yeast, and they are highly specific for those genes, with very few non-RPG binding sites [16,23,24,25,26]. Fork-head transcription factors (also known as winged-helix transcription factors) constitute a family of sequence-specific regulators possessing a conserved DNA-binding domain—a variant of the helix-turn-helix domain. They are important in controlling many cellular and developmental processes in eukaryotes. Coding genes for fork-head transcription factors have been described for budding and fission yeasts, among other organisms. Four genes encoding fork-head transcription factors have been identified in fission yeast: *mei4*, *sep1*, *fkh2*, and *fhl1* [27,28].

The fission yeast *fhl1* gene has not yet been thoroughly characterized. It has been suggested that *fhl1* could regulate sporulation, typically induced under poor nutrient conditions [27]. Deletion of *fhl1* causes a reduced cell growth rate and an extension of cell length due to delayed G2-to-M transition. The mutant deletion strain cells are more resistant to killing by the mutagen methyl methane sulfonate than wild-type cells [27]. The *fhl1* gene is not essential for cell viability, but mutant cells have longer doubling times, and the homothallic cells show stronger sporulation [27,28]. Genome-wide gene expression analysis of the *fhl1* mutant strain has revealed that the expression of several genes is dysregulated. Those dysregulated genes in the mutant strain belong to the nitrogen starvation response, mating, sporulation, nutrient sensing, transport, and permease genes [28]. Tor2 also regulates some of the genes regulated by Fhl1, and it might be possible that Fhl1 acts in TOR signaling downstream of Tor2 [28]. The Fhl1 protein may be regulated by phosphorylation by Tor2 or, alternatively, by a protein kinase regulated by Tor2. Moreover, it has been demonstrated that fission yeast Fhl1 regulates only a few RPGs and stress-response genes, in contrast to the budding yeast *FHL1* gene [28,29,30]. Only two RPGs were found to be downregulated in the *fhl1* mutant strain—namely, *rpl603* and *rps2202*, of which *rps2202* possesses a HomolE box in its promoter. Perhaps the role of Fhl1 is to regulate transcription from HomolE-containing promoters under certain physiological conditions in fission yeast. Notably, promoters containing only a HomolD box have basal transcriptional activity, which could explain why the HomolE box is unnecessary for the RPGs to be transcribed [5,11,18].

The Fhl1 protein has FHA and FH domains, similar to the budding yeast FHL1 protein homolog. The FHA domain might be important to interact with other protein factors involved in transcription, while the FH domain might bind DNA. We did not notice any other important domain in this protein. Additional biochemical work is needed to define the interacting, DNA-binding, and activation domains in the Fhl1 protein. Furthermore, genetic work is necessary to confirm in vivo that the *fhl1* gene regulates the RPGs containing HomolE boxes.

## 4. Materials and Methods

### 4.1. Purification of HomolE-Binding Protein

The HEBP was purified from the wild-type fission yeast strain 972h-. Whole-cell extracts (WCEs) were prepared from 100 g (wet weight) of cells by grinding the cells with a mortar and liquid nitrogen, as described by Rojas et al. [18]. The obtained WCE (100 mL at 10 mg/mL protein concentration) was loaded onto a 50 mL phosphocellulose (Whatman P11) column that had previously been equilibrated with buffer A (20 mM HEPES pH 7.9, 50 mM KCl, 0.5 mM EDTA, 2.5 mM DTT, 0.5 mM PMSF, and 10% *v*/*v* glycerol). After protein loading, the column was washed extensively with buffer A and eluted stepwise with 0.2 M, 0.5 M, and 1.0 M KCl in buffer A. The HEBP activity was determined by electrophoretic mobility shift assay (EMSA), and the binding activity was found in the 0.5 M KCl fraction from the P11 column. This protein fraction (200 mg) was dialyzed against buffer A and loaded onto an S-Sepharose column (20 mL column). Bound proteins were eluted with a linear KCl gradient (0.05 to 0.5 M) in buffer A. Fractions containing HEBP activity were pooled (50 mg) and dialyzed against buffer B (20 mM sodium phosphate pH 7.9, 2.5 mM DTT, 0.5 mM PMSF, and 10% *v*/*v* glycerol) and loaded onto a hydroxyapatite column (10 mL) equilibrated in buffer B. The bound proteins were eluted with a linear gradient of sodium phosphate buffer (0.05–0.25 M) in buffer B. Those fractions containing HEBP activity (5 mg) were pooled and dialyzed against buffer A and loaded onto a Poros-S column (1 mL), and then eluted with a linear gradient of KCl (0.05–0.5 M) in buffer A. HEBP-binding activity was eluted as a sharp peak around a 0.25 M KCl concentration. The fractions containing HEBP activity were pooled and analyzed by SDS-PAGE, followed by silver staining, or analyzed by Southwestern blot, as described below.

### 4.2. Electrophoretic Mobility Shift Assay

EMSA was used to monitor the HEBP activity from the different chromatographic steps. The EMSA was performed as described previously [18]. Each binding reaction contained a binding mix: 20 mM HEPES (pH 7.9), 50 mM KCl, 10 mM MgCl2, 0.1 mM EDTA, 5% glycerol, 0.5% PEG 8000 (Sigma-Aldrich, St. Louis, MO, USA), 5 mM DTT, 0.1 mM PMSF, 5 μg of acetylated BSA, and 0.5 μg of sonicated salmon sperm DNA. The proteins were incubated with the binding mix for 5 min at 25 °C. Then, 5–10 ng of two tandem double-stranded HomolE boxes containing the probe TTCGTTGAGGGTAGGGTTGAGGGTAGGGTTATGC, end-labeled by T4 polynucleotide kinase and ^32^γATP, was added to the assays, and the reaction mixes were incubated for 15 min at 30 °C. The DNA–protein complexes were evaluated in 5% polyacrylamide gels containing 2% glycerol and run at 100 V and 4 °C for 2 h in Tris-borate-EDTA (pH 8.3) buffer. Complex detection was performed by autoradiography analysis.

The EMSA performed with recombinant Fhl1 protein was carried out as described above, with slight modifications. The non-specific competitor salmon sperm DNA was replaced with 50 ng of poly (dG-dC), and the PEG was omitted. The polyacrylamide gels used to resolve the complexes contained 1 mM DTT.

### 4.3. In Vitro Transcription Assays

In vitro transcription was performed as described previously [18], using 100 ng of HomolD-box- or HomolE–HomolD-box-containing templates. The synthetic promoter sequence of the HomolD box (shaded blue) from the rpK5 promoter was AAACAGTCACATTTTACAACAATTCACCACTTCAATTTCCAACCTCAATACCCTATTCTCCAACAACCAA. The synthetic promoter sequence of the HomolE–HomolD construct was CGAATTCGTTGAGGGTAGGGTtgAGGGTAGGGTtgCAGTCACATTTTA CAACAATTCACCACTTCAATTTCCAACCTCAATACCCTATTCTCCAACAACCAA. This promoter contained 2 HomolE boxes (shaded green) separated by a 2 bp spacer (tg). The HomolE boxes are shaded green, and the HomolD box is shaded blue. The mutant HomolE box contained two base changes indicated in lowercase (AGtGTAGtGT). These promoters were fused to the G-minus cassette and, upon digestion of the transcripts by RNase T1, produced a transcript of 370 nucleotides in length. The reactions were performed with 5 μL of WCE (10 mg/mL). Transcript detection was performed by autoradiography analysis.

### 4.4. Southwestern Blot Analysis

Proteins were separated on an 8% SDS-PAGE and electro-transferred to an Immobilon-P membrane in 20 mM Tris, 200 mM glycine, and 20% *v*/*v* methanol. The membrane containing the proteins was incubated overnight at 8 °C in TNED buffer (10 mM Tris pH 7.9, 50 mM NaCl, 0.1 mM EDTA, and 1 mM DTT), supplemented with 2.5% BSA and 20 μg/mL sonicated salmon sperm DNA. Afterward, the membrane was incubated with 1 × 10^6^ cpm/mL of a HomolE end-labeled probe (the same as used for EMSA) or a mutant HomolE end-labeled probe (TTCGTTGAGtGTAGtGTTGAGtGTAGtGTTAT GC) for 14 h at room temperature. After probe incubation, the membrane was washed four times for 20 min with TNED buffer and exposed to an X-ray film.

### 4.5. Recombinant fhl1 Protein Expression and Purification

The coding sequence of the *fhl1* gene (SPAC1142.08) was synthesized at GenScript (Piscataway, NJ, USA) and inserted into the expression vector pET15b (Novagen, St. Louis, MO, USA). *E. coli* BL21 (DE3) cells were transformed with pET15b containing the corresponding gene sequences, grown in LB media until OD_600_ 0.8. Then, the protein expression was induced with 0.5 mM IPTG (Calbiochem, St. Louis, MO, USA) to obtain appropriate amounts of the recombinant protein. To recover the recombinant protein, first, the cell pellet was washed with STE buffer (100 mM NaCl, 10 mM Tris HCl pH 8.0, 1 mM EDTA). Then, the cells were sonicated and centrifuged to collect the pellets, which were washed with lysis buffer (50 mM HEPES pH 7.9, 5% glycerol, 2 mM EDTA, 0.1 mM DTT, 0.05% DOC, 1% Triton X-100) and then washed again with the same buffer but without detergents. The washed pellet was mixed with 40 mL of guanidine buffer (6 M guanidine hydrochloride, 10 mM HEPES pH 7.9, 0.2 mM EDTA, 0.2 mM EGTA, 2 mM DTT) and was incubated overnight at 4 °C. The next day, the protein mix was diluted with 160 mL of dilution buffer (10 mM HEPES pH 7.9, 0.2 mM EDTA, 0.2 mM EGTA, 2 mM DTT) and incubated overnight at 4 °C. Then, the mix was dialyzed for 4 h at 4 °C against 20 mM Tris pH 7.5, 0.5 M KCl, 20 mM imidazole, 10% glycerol, 0.01% Triton X-100, 1 mM β-mercaptoethanol, and 0.1 mM PMSF. The protein mix was centrifuged to discard denatured proteins, and the supernatant was saved for Ni-NTA-agarose resin purification under native conditions. Then, 100 mL of protein mix was passed through 1 mL of settled ProBond Ni-NTA-agarose resin (Invitrogen, Waltham, MA, USA) previously equilibrated with the same dialysis buffer. Then, the resin was washed until no proteins were detected in the flow-through. Afterward, bound Fhl1 protein was recovered via elution with 200 mM imidazole in the dialysis buffer. The purified recombinant Fhl1 protein was dialyzed against 20 mM HEPES pH 7.9, 2.5 mM DTT, 50 mM KCl, 0.1 mM PMSF, 0.5 mM EDTA, and 20% *v*/*v* glycerol. The Fhl1 protein was analyzed with 10% SDS-PAGE, followed by silver staining.

## Figures and Tables

**Figure 1 ijms-24-09472-f001:**
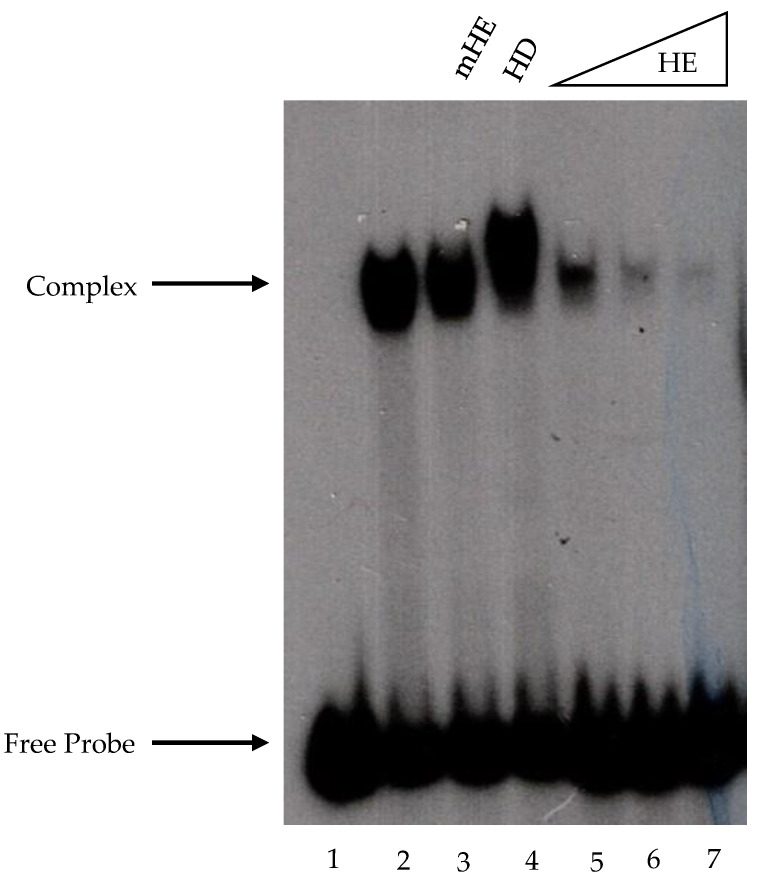
Fission yeast whole-cell extracts contain a HomolE-box-binding protein. Fission yeast whole-cell extracts were prepared as described in the Materials and Methods, and 5 µL of protein (50 μg) was incubated with an end-labeled oligonucleotide containing two HomolE boxes (see Section 4). Complexes were separated by electrophoreses in a 5% polyacrylamide gel in TBE buffer. Lane 1 does not contain protein, and lane 2 contains a fission yeast whole-cell extract. Lane 3 contains a whole-cell extract plus 50 ng of double-stranded cold (unlabeled) mutated HomolE oligonucleotide competitor (mHE; TTCGTTGAGtGTAGtGTTGC), while lane 4 contains 50 ng of double-stranded cold HomolD (HD; TGAAAACAGTCACATTTTAC) oligonucleotide competitor. Lanes 5–7 contain increasing amounts (12.5, 25, and 50 ng, respectively) of double-stranded cold wild-type HomolE (HE; TTCGTTGAGGGTAGGGTTGC) oligonucleotide competitors. Proteins and cold competitors were pre-incubated for 15 min in the reaction buffer before adding the labeled probe. The assay conditions with the cold competitors were similar to the reactions without oligonucleotide competitors.

**Figure 2 ijms-24-09472-f002:**
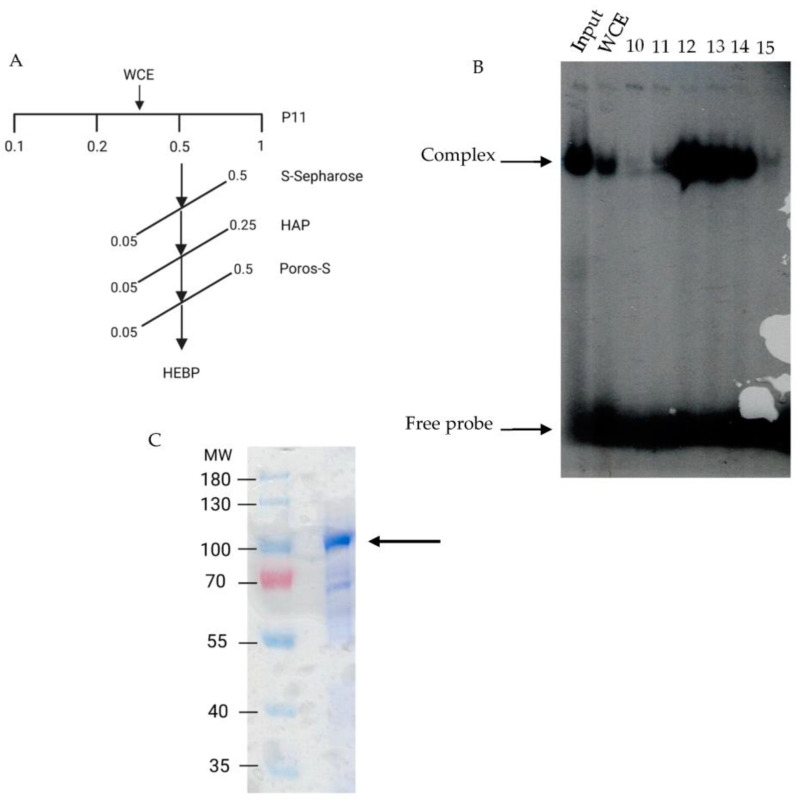
Purification of the HomolE-box-binding protein (HEBP): (**A**) Fission yeast whole-cell extracts were fractionated by chromatography (see Section 4) according to the scheme. (**B**) Electrophoretic mobility assay of the last purification step (Poros-S column), showing the binding activity to the labeled HomolE probe. The Input lane shows the binding activity from the hydroxyapatite column, and the WCE lane shows the binding of the crude whole-cell extract. The column fractions (10–15) were assayed, and those fractions with binding activity (12–14) were pooled and further analyzed. (**C**) SDS-PAGE analysis of the pooled fractions from (**B**). Three micrograms (3 μg) of protein was analyzed in an 8% SDS-PAGE gel followed by Coomassie Blue R-250 staining. The arrow shows the major 100 kDa polypeptide present in the fraction.

**Figure 3 ijms-24-09472-f003:**
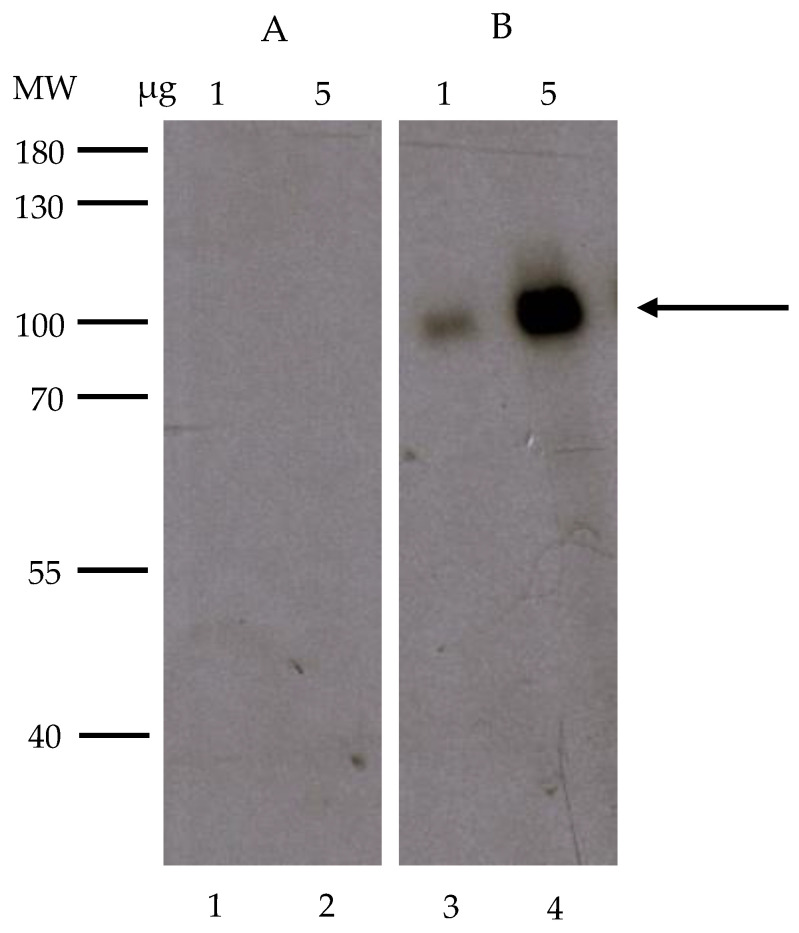
Southwestern blot analysis of the fractions from the Poros-S column. Proteins were separated on an 8% SDS-PAGE gel and transferred to Immobilon-P membranes. After protein renaturation, the membranes were incubated with labeled probes, washed, and exposed to an X-ray film. Panel (**A**) shows the autoradiography of a membrane incubated with a mutant HomolE probe (mHE), while panel (**B**) shows the autoradiography of a membrane incubated with a wild-type HomolE probe (HE). The amounts of loaded protein from the Poros-S column (1 and 5 μg, respectively) are shown at the top. The MW markers are shown on the left side of the figure. The arrow indicates the polypeptide that binds the HomolE box.

**Figure 4 ijms-24-09472-f004:**
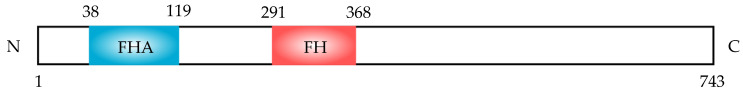
Schematic representation of the Fhl1 protein. The *fhl1* encodes the fission yeast Fhl1 protein, which has 743 amino acids and a calculated molecular weight of 81.13 kDa. The protein possesses a fork-head-associated (FHA, blue) domain (from amino acids 39 to 119) and a fork-head (FH, red) domain (from amino acids 291 to 368).

**Figure 5 ijms-24-09472-f005:**
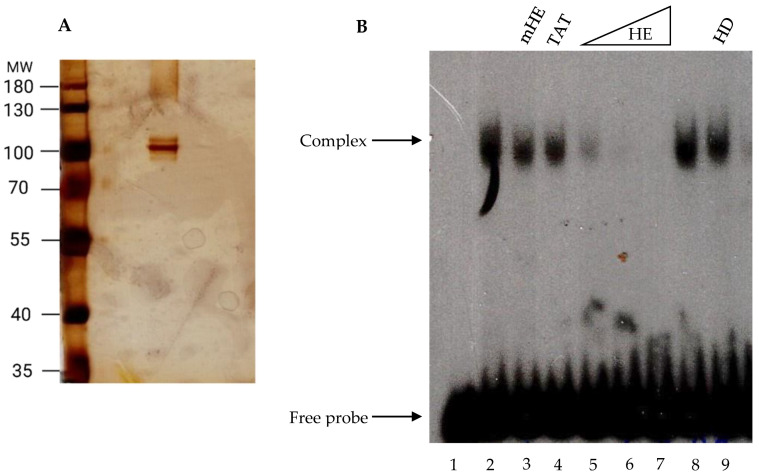
The fission yeast Fhl1 protein binds to HomolE. The gene encoding the fission yeast Fhl1 protein was cloned in a bacterial expression vector, and the recombinant protein was purified by NTA-Ni-agarose affinity chromatography. (**A**) Analysis of 200 ng of the recombinant protein in an 8% SDS-PAGE gel, followed by silver staining. On the left side, the MW markers are shown. The arrow shows the Fhl1 protein. (**B**) Electrophoretic mobility shift analysis of the recombinant fission yeast Fhl1 protein. Assays were performed with 10 ng of the purified protein using the end-labeled HomolE probe and poly (dG-dC) as a non-specific competitor. Lane 1 does not contain protein. Lanes 2 and 8 contain recombinant Fhl1 protein. Lane 3 contains recombinant protein plus 50 ng of cold mutant HomolE (mHE) oligonucleotide, while lane 4 contains recombinant Fhl1 protein plus 50 ng of cold TATA oligo. Lanes 5–7 contain recombinant Fhl1 protein and 12.5, 25, and 50 ng of cold HomolE (HE) oligonucleotide, respectively. Lane 9 contains recombinant Fhl1 protein plus 50 ng of cold HomolD (HD) oligonucleotide. The assay conditions with the oligonucleotide competitors were similar to those described in Figure 1.

**Figure 6 ijms-24-09472-f006:**
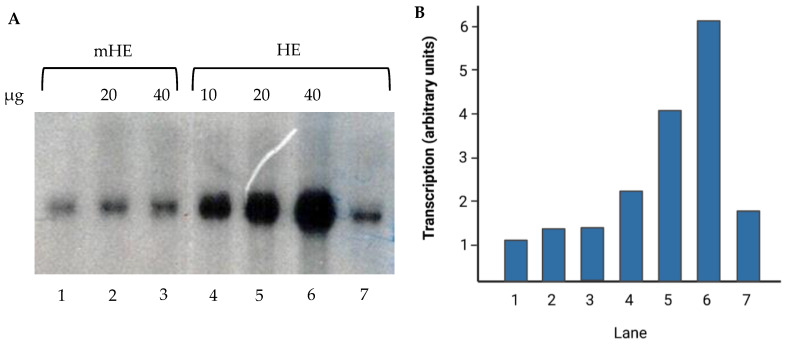
Recombinant Fhl1 protein can activate in vitro transcription: (**A**) Transcription assay using templates containing two mutated HomolE boxes (mHE) or two wild-type HomolE boxes (HE) upstream of a HomolD box. Transcription assays were carried out using 5 µL (50 μg) of fission yeast WCE. Lanes 1–3 contain the mHE promoter. In lane 1, the Fhl1 protein was not added, while lanes 2 and 3 contain 20 and 40 ng of recombinant Fhl1 protein, respectively. Lanes 4–7 contain the HE promoter, and lanes 4–6 contain 10, 20, and 40 ng of recombinant Fhl1 protein, respectively. Lane 7 does not contain recombinant Fhl1 protein. (**B**) A plot displaying the quantification from (**A**).

## Data Availability

All data are shown within the manuscript.

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
