# Peer review of "The Product of the Fission Yeast fhl1 Gene Binds to the HomolE Box and Activates In Vitro Transcription of Ribosomal Protein Genes"

_ijms, 2023, doi:10.3390/ijms24119472_

Round 1

Reviewer 1 Report

In this paper, the fission yeast Fhl1 protein was characterized with respect to its binding ability to a promoter element termed HomolE box. Homol E boxes are found in promoters of ribosomal protein coding genes. The biochemical assays presented in this manuscript are convincing and in vitro transcription activation by Fhl1 was demonstrated. Previously published in vivo data for a fission yeast fhl1 mutant strain could be discussed in some more detail.

The authors assume that Fhl1 is responsible for the observed reduction in electrophoretic mobility shown in Figure 1. This assumption could be easily tested by EMSA assay of protein extract from a fhl1 mutant strain, and it would strongly support the authors interpretation. If such approach can’t be included, this option could at least be mentioned for future verification.

Fhl1 was previously shown to regulate ribosomal and stress responsive genes https://doi.org/10.1007/s00294-016-0607-1 (cited in the manuscript). In this paper fhl1 mutants were phenotypically characterized and transcriptome alterations detected. Apparently, some of the ribosomal genes are deregulated in the mutant. Based on the identification of Fhl1p as a HomolE binding protein, promoters of known fhl1 regulated genes should be checked for potential HomolE-like binding sites and results mentioned in the discussion.

Do the established phenotypes for fhl1 mutants fit into the assumption that Fhl1 might be an HomolE binding protein relevant for expression of ribosomal protein genes?

A brief introduction into the molecular function of forkhead transcription factors might be included

The EMSA assay was conducted in the presence of an unlabeled mutated HomolE box competitor. The method part for this assay should be extended to include the sequences for the mutated HomolE and HomolD oligonucleotide competitors. Assay conditions with the competitors were presumably similar to the reaction without, please add a brief statement.

Please verify that conventions for writing fission yeast and budding yeast genes, mutant genes and proteins are followed in the entire manuscript

Author Response

First, we would like to thank both reviewers for their valuable comments on the manuscript. Those comments will undoubtedly improve the quality of it. Second, enclosed are the answers to the reviewer’s comments and suggestions. We hope that now will be ready for publication in IJMS.

Reviewer 1

  1. We have discussed in some more detail the previously published data on the fhl1 mutant deletion strain.

  1. Thank you for suggesting the EMSA experiment with extracts from the fhl1 mutant strain. Unfortunately, our laboratory does not have the deletion mutant strain. In the near future, we plan to get that strain from Dr. Miklos in Hungary or the Bioneer Consortium (https://us.bioneer.com/products/spombe). We will conduct those experiments as soon as we get the mutant strain.

  1. Thank you for suggesting checking for HomolE boxes in the promoter genes regulated by Fhl1p. Indeed, few ribosomal and stress-responsive genes are dysregulated in the fhl1 mutant strain. We checked the promoters of those for HomolE boxes using the Eukaryotic Promoter Database and Pombase. Only two RPGs were reported that were downregulated in the mutant fhl1 strain. Those are rpl603 and rps2202. Only the promoter of rps2202 contains a HomolE box. However, in fission yeast, each ribosomal protein is encoded by a gene family, and in each family, not all the members contain a HomolE box; therefore, each member containing a HomolE box should be individually examined in the fhl1 mutant strain to check for dysregulation of those genes.

  1. Yes, the phenotype for fhl1 mutant cells might fit into the assumption that Fhl1p may be relevant for the expression of RPGs since those mutant cells show a slow-growth phenotype. Additionally, it interacts with the TORC1 pathway, which is important in controlling protein synthesis. Moreover, database mining (Pombase) indicates that the fhl1 gene in fission yeast has negative genetic interactions with some RPGs, such as rps23 and rps402, which contain HomolE boxes at their promoters.

  1. A brief paragraph on the fork-head transcription factors was added in the Discussion section.

6. We included the sequences of the mutated HomolE and Homol D boxes competitors in the Figure 1 legend and indicated that all reactions had similar assay conditions.

  1. Yes, we checked the conventions for writing fission and budding yeast genes, mutant genes, and proteins were the same and followed along the entire manuscript. Thank you very much for the suggestion.

Reviewer 2 Report

In the manuscript entitled "The product of the fission yeast fhl1 gene binds to the HomolE box and activates in vitro transcription of ribosomal protein genes" authors purify the protein able to specifically bind HomolE element, they then test whether the specific protein encoded by fhl1 gene bind the sequence and stimulates the transcription, which they confirm. The experiments are adequately designed, executed and interpreted. I do not have any major concerns.

My only minor concern is English in the Introduction section. (see below)

The manuscript is written in good English and can be easily followed. However there are parts in the Introduction, in which English should be edited. Namely these are:

First and second paragraph - minor revision of grammar.

Fourth paragraph - second half of the paragraph with a part about inverted arrangements can be improved to be more easy to follow.

These are, however, only suggestions. 

Author Response

First, we would like to thank both reviewers for their valuable comments on the manuscript. Those comments will undoubtedly improve the quality of it. Second, enclosed are the answers to the reviewer’s comments and suggestions. We hope that now will be ready for publication in IJMS.

Reviewer 2

Thank you very much for your suggestions. The English language has been edited in those indicated sentences for better understanding.